# ABA Inhibits Rice Seed Aging by Reducing H_2_O_2_ Accumulation in the Radicle of Seeds

**DOI:** 10.3390/plants13060809

**Published:** 2024-03-12

**Authors:** Qin Zheng, Zhenning Teng, Jianhua Zhang, Nenghui Ye

**Affiliations:** 1College of Agronomy, Hunan Agricultural University, Changsha 410128, China; zhengqin@stu.hunau.edu.cn (Q.Z.); sailingtzn@163.com (Z.T.); 2School of Life Sciences and State Key Laboratory of Agrobiotechnology, The Chinese University of Hong Kong, Shatin 999077, Hong Kong; 3Shenzhen Research Institute, The Chinese University of Hong Kong, Shenzhen 518057, China; 4Department of Biology, Hong Kong Baptist University, Kowloon 999077, Hong Kong

**Keywords:** rice, seed aging, ABA, H_2_O_2_, germination, seedling establishment

## Abstract

The seed, a critical organ in higher plants, serves as a primary determinant of agricultural productivity, with its quality directly influencing crop yield. Improper storage conditions can diminish seed vigor, adversely affecting seed germination and seedling establishment. Therefore, understanding the seed-aging process and exploring strategies to enhance seed-aging resistance are paramount. In this study, we observed that seed aging during storage leads to a decline in seed vigor and can coincide with the accumulation of hydrogen peroxide (H_2_O_2_) in the radicle, resulting in compromised or uneven germination and asynchronous seedling emergence. We identified the abscisic acid (ABA) catabolism gene, *abscisic acid 8′-hydroxylase 2* (*OsABA8ox2*), as significantly induced by aging treatment. Interestingly, transgenic seeds overexpressing OsABA8ox2 exhibited reduced seed vigor, while gene knockout enhanced seed vigor, suggesting its role as a negative regulator. Similarly, seeds pretreated with ABA or diphenyleneiodonium chloride (DPI, an H_2_O_2_ inhibitor) showed increased resistance to aging, with more robust early seedling establishment. Both *OsABA8ox2* mutant seeds and seeds pretreated with ABA or DPI displayed lower H_2_O_2_ content during aging treatment. Overall, our findings indicate that ABA mitigates rice seed aging by reducing H_2_O_2_ accumulation in the radicle. This study offers valuable germplasm resources and presents a novel approach to enhancing seed resistance against aging.

## 1. Introduction

Seeds play a critical role in the plant’s life cycle, serving as essential materials for planting and crop production in subsequent seasons. Therefore, it is imperative for seeds to exhibit high yield potential and produce viable and vigorous offspring [1]. Following harvest, crop seeds are typically stored under ambient conditions for varying durations, ranging from weeks to years [2]. The germination process plays a pivotal role in assessing the viability and vigor of seeds, thereby contributing to successful crop establishment in the soil [3]. Seed vigor, which influences the rapid and uniform emergence of the radicle, stands as a critical factor in this process. The longevity of seeds determines the vigor index, which is influenced by their physiological characteristics, genetic composition, and the rate of deterioration during storage [4]. Additionally, various environmental factors such as temperature, humidity, moisture content, oil content, pathogen attacks, mechanical damage, storage duration, and gaseous exchange directly or indirectly impact seed vigor [5].

The accumulation of reactive oxygen species (ROS) and ABA has been widely recognized as a key factor contributing to seed deterioration during ambient storage [6,7,8,9,10]. The interplay between ABA and ROS has been linked to plant growth, development, and stress responses [8,10,11,12,13,14,15,16,17]. H_2_O_2_ mediates ABA-induced plant stomatal closure and tolerance to heat stress [14,15]. H_2_O_2_ influences the regulation of ABA catabolism during seed imbibition, thereby modulating seed dormancy and germination [13]. The accumulation of H_2_O_2_ induced by ABA may also participate in the early induction of various antioxidative genes, thereby enhancing tolerance to cold stress [16]. In rice seeds, ABA decreases ROS production, leading to the repression of ascorbate and gibberellin accumulation [17]. Overproduction of ROS or hormonal imbalances can lead to disruptions in growth enzymes, metabolism, and alterations in cellular membranes and the cytoplasmic state during storage [18,19,20]. However, whether and/or how the control of seed aging by ABA was connected with ROS remains unclear.

Several methods are being used to improve seed resistance against aging. Genetic improvement is one approach. For example, ectopic expression of *NnPER1*, a *Nelumbo nucifera* 1-cysteine peroxiredoxin antioxidant in *A. thaliana* enhanced seed vigor and longevity [21]. In rice, the overexpression of *PER1A* improved resistance against aging, while knockout of *PER1A* increased the rate of seed-viability loss during aging compared to the wild type [8]; Physicochemical methods are another approach. For instance, soybean seeds stored at 10 °C and relative humidity below 40% can be preserved for up to two seasons [22]. Priming seeds prior to aging treatment, which involves mixing seeds with moist vermiculite at 25 °C for 36 h followed by air-drying to the original moisture level, partially overcomes the deteriorative effects of aging and improves seed quality [23]. Nanotechnology can also be employed for the rejuvenation of aged seeds [24]; however, more studies are needed to find effective methods to prevent seed aging. In this study, we showed that age-induced H_2_O_2_ accumulation in the radicle was identified as the primary cause of compromised or uneven germination and asynchronous seedling emergence. Additionally, we demonstrated that *OsABA8ox2*, acting as a negative regulator of seed vigor, can be utilized for breeding aging-resistant varieties.

## 2. Results

### 2.1. Aging Significantly Decreases Rice Seed Germination and Seeding Establishment

The impact of aging on rice seed germination and seedling establishment was evaluated by subjecting seeds to various aging durations (i.e., 24 h, 48 h, 72 h). Compared to seeds without aging treatment, those subjected to storage treatments exhibited significantly lower germination rates and compromised germination status, with inhibited radicle and coleoptile growth observed with increasing aging time (Figure 1A,B). While unaged seeds achieved nearly 100% germination, those aged for 24 h and 48 h displayed final germination rates of approximately 60% and 40%, respectively, with almost complete failure of germination observed in seeds aged for 72 h (Figure 1A,B). Analysis using 2,3,5-triphenyltetrazolium chloride (TTC) staining showed weaker staining in the coleorhiza, radicle, and coleoptile of rice seeds aged for 48 h compared to the CK, indicating reduced embryo viability, particularly in the radicle (Figure 1C).

Furthermore, equal numbers of rice seeds subjected to varying aging times were planted in soil, and the percentage of seedling establishment was recorded. Results indicated significantly lower seedling establishment rates for seeds aged for 24 h and 48 h compared to the control, with notably suppressed radicle growth observed (Figure 1D,E). These findings confirm that artificial aging significantly diminishes radicle viability and adversely affects rice seed germination and seedling establishment. Notably, the radicle appears to be more sensitive to aging, experiencing greater damage compared to the coleoptile.

### 2.2. More H_2_O_2_ Is Accumulated in the Radicle of Seeds under Aging Treatment

As reactive oxygen species (ROS), particularly H_2_O_2_, are commonly recognized as crucial factors influencing seed longevity [8,25], we investigated H_2_O_2_ accumulation to evaluate overall oxidative stress in aged rice seeds (Figure 2). Remarkably, the embryos of seeds aged for 48 h exhibited more intense staining than the CK, particularly in the radicle (Figure 2A). Additionally, we conducted a relative quantification of H_2_O_2_ levels in seed embryos. The results revealed significantly higher H_2_O_2_ contents in the embryos of aged seeds compared to the CK during accelerated aging (Figure 2B). This finding corroborates the observations from DAB staining and provides insight into the radicle’s demise.

### 2.3. Inhibition of ABA Degradation Increases Germination and Seedling Establishment from Aged Rice Seeds

The plant hormone ABA plays a crucial role in regulating seed vigor [8,9,26]. According to transcriptomic data reported by Wang et al. [8], the ABA catabolism gene, *abscisic acid 8′-hydroxylase 2* (*OsABA8ox2*), which is significantly induced by aging treatments, appears to play an important role in ABA catabolism during the artificial aging process. To investigate whether OsABA8ox2 influences seed aging, we generated rice *OsABA8ox2* knockout mutants using the CRISPR/Cas9 method as in [27] and an *OsABA8ox2* overexpression line. Sequencing analysis confirmed the presence of two independent mutants, KO-3 and KO-11, which harbored a 1-bp deletion and a 1-bp insertion in the first exon of *OsABA8ox2*, respectively (Figure 3A,B). Subsequently, we assessed the ABA content in rice seeds of the *OsABA8ox2*-OE line and *OsABA8ox2* mutants. Our results revealed that overexpression of *OsABA8ox2* significantly reduced the ABA content in rice seeds, while the content in *OsABA8ox2* mutants increased significantly (Figure 3C).

To evaluate the impact of *OsABA8ox2* on seed aging, we conducted an artificial aging test. Our observations showed that overexpression of the *OsABA8ox2* gene led to a dramatic reduction in seed germination completion in accelerated aging (Figure 3D,E). Notably, seeds of the *OsABA8ox2*-OE line exhibited mortality during maturation and after drying, in contrast to knockouts of the *OsABA8ox2* gene (KO-3 and KO-11), which demonstrated substantially improved seed resistance against aging. Specifically, after aging for 48 h, over 80% of *OsABA8ox2* mutants remained viable compared to only 43% for the wild type (ZH11). Furthermore, while all ZH11 seeds had perished following 72 h of aging treatment, approximately 45% of *OsABA8ox2* mutant seeds were still viable (Figure 3E). Additionally, *OsABA8ox2* mutant seedlings exhibited superior growth compared to the ZH11 seedlings derived from rice seeds aged for 48 h (Figure 3F). These findings support the conclusion that inhibiting ABA degradation enhances seed resistance against aging and improves seedling establishment from aged rice seeds.

### 2.4. Exogenous ABA Reduces the Concentrations of H_2_O_2_ and Positively Regulates the Seedling Establishment in Artificially Aged Rice Seeds

The interplay between ABA and H_2_O_2_ in plant tissues has been extensively discussed in the literature [11,28], with confirmation of their pivotal roles in seed vigor [8]. Thus, we investigated whether the increased resistance against aging observed in *OsABA8ox2* mutants is associated with changes in H_2_O_2_ and ABA levels. To test this hypothesis, we examined H_2_O_2_ contents in *OsABA8ox2* mutant seeds and observed that the H_2_O_2_ levels in seeds of KO-3 were significantly lower than those of the WT (Figure 4A,B). We postulated that the existence of ABA reduces the concentrations of H_2_O_2_ during accelerated aging. Indeed, seeds primed with ABA exhibited a slight, albeit not significant, increase in H_2_O_2_ content after 48 h of aging treatment (Figure 4A,B). Similarly, seeds pretreated with DPI also showed reduced H_2_O_2_ accumulation in the embryo following aging treatment (Figure 4A,B).

Consistent with *OsABA8ox2* mutant seeds, seeds pretreated with ABA and DPI exhibited higher survivability than the CK during accelerated aging treatment, particularly notable when aged for 72 h, with over 30% survival compared to none in the CK group (Figure 4C). Moreover, the germination rates, radicle growth, and early seedling establishment of seeds pretreated with ABA and DPI were comparable to those of unaged ZH11 seeds (Figure 4D,E). These findings collectively suggest that a low concentration of ABA reduces H_2_O_2_ concentrations and positively regulates seedling establishment in artificially aged rice seeds. Furthermore, this study offers novel insights and strategies for breeding and genetic enhancement aimed at improving seed storage longevity and aging tolerance.

## 3. Discussion

Seed aging represents an irreversible process characterized by the gradual decline of seed vigor, including reduction in antioxidant systems, disruption of cellular membranes, genetic integrity damage, lipid peroxidation, and protein degradation [29,30,31,32]. ROS serve as key modulators of seed aging, initiating early bursts during accelerated aging, potentially leading to dynamic changes in mitochondria and seed inactivation [8,33]. The plant hormone ABA plays an indispensable role in regulating seed maturation and vigor [8,9,26]. Several studies have explored strategies to slow down seed aging and enhance seed vigor by manipulating ABA and H_2_O_2_ synthesis, degradation, and downstream signaling pathways [8,21,34,35]. Under artificial aging conditions, rice varieties with higher levels of ABA and lower levels of H_2_O_2_ demonstrate enhanced resistance to aging [8]. In this study, we provide new insights into the crosstalk between ABA and H_2_O_2_ during seed aging and offer novel insights and novel approaches to enhance seed resistance against aging.

Prolonged storage under high-temperature and high-humidity conditions results in reduced seed quality and vigor, making seeds less viable after sowing [36,37,38,39]. Similarly, our results demonstrate that seed aging during storage leads to decreased germination and loss of seed vigor, which is irreversible (Figure 1). During accelerated aging, aged seeds initially display suppressed germination, followed by radicle viability loss, and ultimately complete inactivation (Figure 1A–C). This underscores the radicle’s heightened sensitivity to aging stress, experiencing greater damage than the coleoptile. Histochemical staining and H_2_O_2_ quantification in aged rice seeds provide insights into this phenomenon, with increased H_2_O_2_ accumulation observed in the embryo, particularly in the radicle (Figure 2). This research reinforces the potential of ROS scavenging to enhance seed resistance against aging [21,40,41].

Our findings suggest that the regulation of seed vigor by H_2_O_2_ in rice seeds may be linked to the ABA degradation pathway, a cross-talk often associated with plant growth, development, and stress responses [8,10,11,12,13,14,15,16,17]. Under aging treatment, *OsABA8ox2* was significantly induced, leading to decreased ABA content [8] and increased H_2_O_2_ content (Figure 2). Subsequent validation of *OsABA8ox2* functionality through transgenic lines revealed that overexpression of *OsABA8ox2* reduced ABA content and conferred aging tolerance to transgenic seeds. Conversely, seeds of *OsABA8ox2* mutants or those pretreated with ABA and DPI exhibited lower H_2_O_2_ levels and increased resistance to aging, resulting in more robust early seedling establishment (Figure 3). Overall, these findings suggest that manipulating H_2_O_2_ scavenging, particularly in the radicle, and modulating the ABA degradation pathway through *OsABA8ox2* represent effective strategies to enhance seed resistance against aging. In addition, pretreatment with exogenous ABA or ROS scavengers is a simple and effective method to enhance seed resistance to aging.

Some factors in the ABA signaling pathway have been demonstrated to participate in seed aging [34,42,43]. For example, the extreme alleles of *abscisic acid insensitive 3* (*ABI3*), a crucial downstream component of ABA signaling, have been found to severely compromise seed longevity and vigor, while the ABA biosynthesis mutant *abscisic acid deficient 1* also exhibits reduced traits [42,43]. Additionally, the bZIP transcription factor encoded by *ABSCISIC ACID INSENSITIVE 5* (*ABI5*) serves as a key regulator of seed maturation and vigor in legumes [34]. These findings, along with those of the present study, significantly enhance our understanding of the molecular mechanisms underlying seed-aging regulation. However, identifying the specific ABA signal and elucidating the mechanism by which ABA regulates seed aging through the ROS signaling pathway are key areas for our future research.

## 4. Materials and Methods

### 4.1. Plant Materials and Growth Conditions

Rice variety ZH11 (*O. sativa* cv. Zhong Hua 11) was used in this study. *OsABA8ox2* mutants KO-3 and KO-11 and overexpression line OE-26 were as reported previously [27]. All varieties were cultivated in paddy fields at Hunan Agricultural University during the rice-growing season and were harvested simultaneously 40 days after flowering, then dried in sunlight for one week. Afterward, the rice seeds underwent an accelerated seed-aging test following a three-month equilibration period at room temperature. Upon germination, all germinated rice seeds were transferred to soil and cultivated in greenhouses under a photoperiod of 16 h light and 8 h dark at 25 °C. After one week, the rates of seedling establishment were quantified as the percentage of germinated seeds that successfully developed into established seedlings.

### 4.2. Accelerated Seed-Aging Test

The accelerated seed-aging procedure followed the method outlined by Wang et al. [8]. Dry seeds were subjected to aging in a tightly closed container maintained at 100% relative humidity and a temperature of 45 °C. For germination analysis, a minimum of 25 aged seeds were selected and placed on moistened filter paper in darkness at 30 °C. Germination was scored based on a ≥1 mm coleoptile emergence. The germination rates of seeds were recorded daily. Germination ratio (%) = number of germinated seeds/total number of seeds × 100%. Each treatment was replicated three times, and the data were subjected to statistical analysis.

### 4.3. Histochemical Staining and Quantification of H_2_O_2_

Aged seeds were imbibed in 0.1% 3,3′-Diaminobenzidine (DAB; adjusted to pH 6.0 with HCl) for 3 h at 25 °C in darkness. Afterward, aged seeds were rinsed three times with deionized water, and the staining was scanned using an EPSON V700 Scanner (EPSON (China) Co., Ltd., Beijing, China). H_2_O_2_ levels were measured using H_2_O_2_ reagent kits (Beijing Solarbio Science & Technology Co., Ltd., Beijing, China) following the manufacturer’s instructions. A total of 25 embryos (about 20 mg) were homogenized and mixed in 500 μL lysis buffer. After centrifuging for 5 min at 4 °C at 12,000× *g*, 250 μL of supernatant was blended and homogenized with 25 μL detection reagent 2 and 50 μL detection reagent 3. After centrifuging for 5 min at 4 °C at 12,000× *g*, the supernatant was discarded, and 250 μL detection reagent 4 was added to the precipitate. After dissolution, it was used to detect H_2_O_2_ levels. H_2_O_2_ was detected at 415 nm and the absolute amount was calculated based on a standard curve.

### 4.4. ABA and DPI Pretreatments

For seed pretreatment method, seeds were initially surface-disinfected with 5% NaClO and subsequently rinsed multiple times with deionized water. Seeds were then immersed in 10 μM ABA or 20 μM DPI for a duration of 4 h at 4 °C. Seeds that were soaked solely in deionized water served as the control (CK). After pretreatment, the seeds were air-dried in a shaded area at room temperature, then sealed in polyethylene bags, and stored at 4 °C for further experimentation.

### 4.5. ABA Measurement

The quantification of ABA contents was conducted by Wuhan MetWare Biotechnology Co., Ltd. (Wuhan, China), as previously described [27]. Briefly, fifty seeds were rapidly frozen in liquid nitrogen and ground into a fine powder. Subsequently, 50 mg of the powdered sample was utilized for hormone extraction, using a mixture of methanol, water, and formic acid in the ratio of 15:4:1. The combined extracts were then evaporated to dryness under a stream of nitrogen gas and redissolved in 80% methanol. The resulting sample extracts were filtered and analyzed using an LC-ESI-MS/MS system (consisting of an HPLC Shim-pack UFLC SHIMADZU CBM30A system coupled with an MS Applied Biosystems 6500 Triple Quadrupole). The internal standards of 2H6-ABA were employed in this study for accurate quantification.

### 4.6. Seed Viability

The viability of seeds was estimated using 2,3,5-triphenyltetrazolium chloride (TTC) staining. For TTC staining, after incubation in water for 12 and 48 h, 30 aged and unaged seeds were selected and stained in 0.5% TTC and kept in the dark at 35 °C for 3 h, and then washed three times with water. Subsequently, the seeds were scanned using an EPSON V700 Scanner.

## Figures and Tables

**Figure 1 plants-13-00809-f001:**
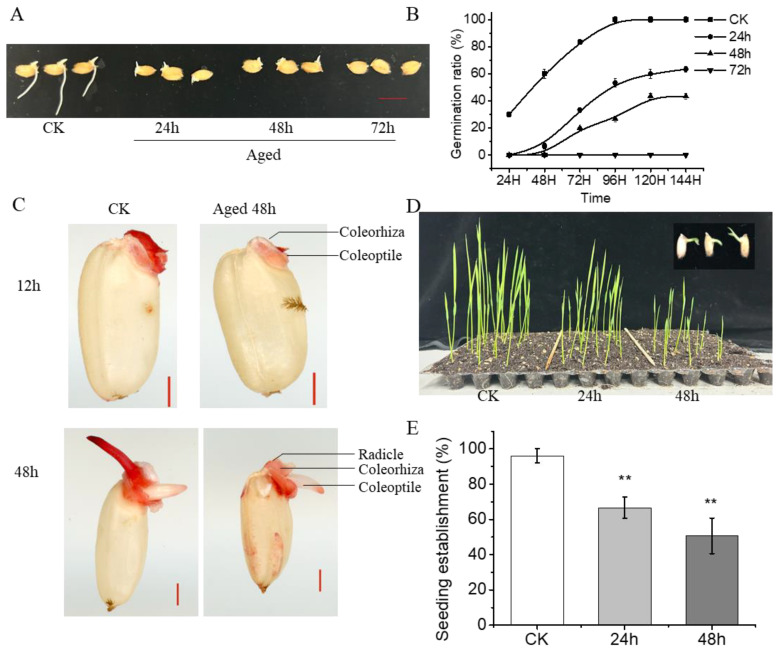
Aging significantly decreases rice seed germination and seedling establishment abilities. (**A**) Representative photographs of aged rice seeds during the imbibition process (48 h after sowing). Rice seeds were aged for 24, 48 or 72 h, 3 months after harvest, and then subjected to analysis. Scale bar = 10 mm. (**B**) Germination rates of rice seeds after different aging treatments. (**C**) 2,3,5-triphenyltetrazolium chloride (TTC) staining of CK and seeds aged for 48 h. Scale bar = 1 mm. (**D**) The early seedling establishment phenotype of different samples (**A**) immediately following germination (1 week after sowing). The top right corner of the image shows rice seeds subjected to 48 h of aging treatment, which fail to develop normal radicles and roots during early seedling growth. (**E**) The quantitative analysis of the seedling establishment rates of different samples are shown (1 week after sowing). Percentages are the average of three repeats ± SE. The asterisk (**) indicates a significant difference at *p* < 0.01 by Student’s *t*-test analysis.

**Figure 2 plants-13-00809-f002:**
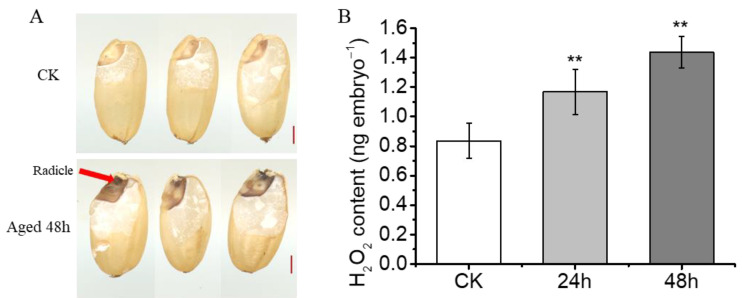
Histochemical staining and quantification of H_2_O_2_ in aged rice seeds. (**A**) In situ detection of H_2_O_2_ during seed aging using 3,3′-Diaminobenzidine (DAB) staining agents. Arrow indicates the production site of H_2_O_2_. Scale bar = 1 mm. (**B**) Quantification of H_2_O_2_ content during seed aging. Values are the average of three repeats ± SE. The asterisk (**) indicates a significant difference at *p* < 0.01 by Student’s *t*-test analysis.

**Figure 3 plants-13-00809-f003:**
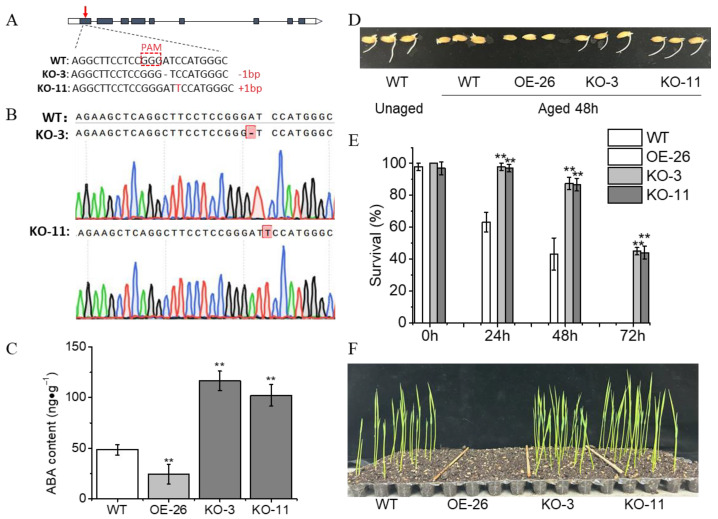
*OsABA8ox2* mutant seeds showed stronger aging resistance, and their seedlings proved to be more robust during early seedling establishment. (**A**) The gene structure of *OsABA8ox2* and the mutated sequence of *OsABA8ox2* mutants produced via CRISPR-Cas9. KO-3 harbors a 1-bp deletion (dashed line) and KO-11 contains a 1-bp insertion (red letter). PAM, protospacer adjacent motif. The red arrow indicates the mutation site. (**B**) Chromatogram of the *OsABA8ox2* sequence across the mutated site in WT and the *OsABA8ox2* mutants. Red box shows the base insertion or deletion in *OsABA8ox2*. (**C**) ABA content of rice seeds of the *OsABA8ox2*-OE line and *OsABA8ox2* mutants. (**D**) Representative photographs of aged rice seeds (**C**) during the imbibition process (48 h after sowing). (**E**) Survival rate of the *OsABA8ox2*-OE lines and *OsABA8ox2* mutant seeds after aging treatments. (**F**) The early seedling establishment phenotype of different samples (**D**) immediately following germination (1 week after sowing). After 48 h of aging treatment, seedlings fail to establish from the OE-26 seeds. Values are the average of three repeats ± SE. The asterisk (**) indicates a significant difference at *p* < 0.01 by Student’s *t*-test analysis.

**Figure 4 plants-13-00809-f004:**
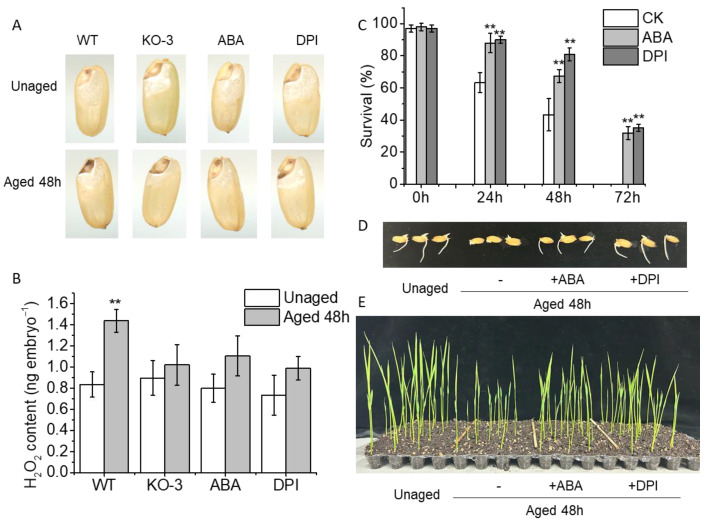
ABA and diphenyleneiodonium chloride (DPI) treatments reduce H_2_O_2_ content and positively regulate the seedling establishment in artificially aged rice seeds. (**A**) Histochemical staining with DAB to determine the localization of H_2_O_2_ in aged rice seeds. (**B**) Quantification of H_2_O_2_ content after seed aging (48 h after aging treatment). (**C**) Survival rate of rice seeds pretreated with ABA or DPI under different aging treatments. (**D**) Representative photographs of aged rice seeds (**C**) compared with ZH11 without aging treatment during the imbibition process (48 h after sowing). (**E**) The early seedling establishment phenotype of different samples (**D**) immediately following germination (1 week after sowing). Values are the average of three repeats ± SE. The asterisk (**) indicates a significant difference at *p* < 0.01 by Student’s *t*-test analysis.

## Data Availability

Data are contained within the article.

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
