# Peer review of "ABA Inhibits Rice Seed Aging by Reducing H2O2 Accumulation in the Radicle of Seeds"

_plants, 2024, doi:10.3390/plants13060809_

Round 1

Reviewer 1 Report

Comments and Suggestions for Authors

The manuscript by Zheng et al presents a study on seed aging in rice. The authors show that seed aging is reduced by lowering the accumulation of hydrogen peroxide in the seed radicle. The experimental system used seeds treated to accelerate aging by incubating at 45 oC and 100% humidity for 24, 48 or 72 h. ABA and reactive oxygen species are known to be major players in seed deterioration. This study presents supportive evidence that the OsABA8ox2 encoding the ABA catabolic enzyme abscisic acid 8’-hydroxylase 2 in seed aging.  Steady state levels of OsABA8ox2 transcripts are enhanced by the aging treatment. Moreover, knockout mutants and overexpressor display enhancement and reduction of seed vigor, respectively, suggesting that OsABA8ox2 acts as a negative regulator of seed aging. Moreover, aging seeds of the OsABA8ox2 mutants accumulated less H2O2 than that of WT.  

Generally, the manuscript can be accepted for publication upon answering the following points: 

Majot point 

  1. Line 146, 148. The authors show that H2O2 is accumulated in aging seeds. Accumulation may result by enhanced synthesis or reduced degradation. As they did not assay these options, they cannot phrase that “H2O2 is produced”.  

  1. Fig. 3. B. this reviewer does not see that chromatograms correlated with the presented nucleotide sequences. 

  1. Fig. 3 A and B. The DNA sequences shown in A and B are different although they claim to present the site of the mutations.  

  1. Fig. 3C. The authors used only one OE line. Do they have additional independent lines to demonstrate that the phenotype results from overexpressing the transgene and not from positional effect of its insertion? 

  1. The authors assayed germination but present the germination data in diverse ways: Fig. 1Bratio (out of 1), Fig. 1E, % (0ut of 100); Fig. 2E, and 4E- survival rather than germination. This is very confusing. Please use a single option.  

  1. Line 202. The author used a single ABA concentration but argued that “a certain amount of ABA...”. This is not supported by any data. 

  1. Fig. 4A. The author does not present data for KO-11.  

  1. Seed presented at Fig. 3D and 4D (unaged and aged WT) are the same picture.  

  1. WT is presented as WT (for example Fig. 3C) or ZH11 (for example Fig. 3D). This is very confusing.  

Minor points 

  1. H2O2 should be printed where the digits are in subscripts. 

  1. Line 22. As the abstract should be understood by its own, please add the role of DPI. 

  1. Line 51. Spell out the full name of the PER1 gene. 

  1. Line 52. Both species names should be in Italics. 

  1. Line 86-7. Please indicate the ratio of embryo (in weight or number) to buffer used. 

  1. Line 87. Change “lysate” to lysis buffer. 

  1. Line 88. Did you homogenize the supernatant with H2O2, or mixed it by using a vortex? 

  1. Line 88. Please indicate the concentration of the added H2O2 solution used. 

  1. Line 94. Rephrase to “immersed in 10 mM ABA or 20 mM DPI”  

  1. Line 140 and Figure 1b. The germination rate is presented as %. You show a fraction of germinated seeds (out of 1). 

  1. Fig. 2B. The units of H202 (ng embryo-1) are not clear. Is that ng H202 per a single embryo? 

  1. Linev199 – is 1 week (not 1 weeks) after sowing a “immediately following germination”? 

  1. Fig. 4 D and E. Indicate in the legend what “CK’ is, or change the test under the panel to “ (empty) , +ABA, +DPI” 

  1. References. Please correct styles: species names in Italics. For example, see lines 288 and 299. 

  1. References, Check journal names. The first letter should be in CAPS. For example, see line 312 and 317. 

Author Response

Reviewer 1:The manuscript by Zheng et al presents a study on seed aging in rice. The authors show that seed aging is reduced by lowering the accumulation of hydrogen peroxide in the seed radicle. The experimental system used seeds treated to accelerate aging by incubating at 45 oC and 100% humidity for 24, 48 or 72 h. ABA and reactive oxygen species are known to be major players in seed deterioration. This study presents supportive evidence that the OsABA8ox2 encoding the ABA catabolic enzyme abscisic acid 8’-hydroxylase 2 in seed aging.  Steady state levels of OsABA8ox2 transcripts are enhanced by the aging treatment. Moreover, knockout mutants and overexpressor display enhancement and reduction of seed vigor, respectively, suggesting that OsABA8ox2 acts as a negative regulator of seed aging. Moreover, aging seeds of the OsABA8ox2 mutants accumulated less H2O2 than that of WT.  

Generally, the manuscript can be accepted for publication upon answering the following points: 

Majot point 

  1. Line 146, 148. The authors show that H2O2 is accumulated in aging seeds. Accumulation may result by enhanced synthesis or reduced degradation. As they did not assay these options, they cannot phrase that “H2O2 is produced”.

Response to comments:

Many thanks for the comment. Yes, it does. We have replaces “produced” and “production” by “accumulated” or “accumulation”.

  1. 3. B. this reviewer does not see that chromatograms correlated with the presented nucleotide sequences.

Response to comments:

We are very sorry for the mistake. We have presented the sequence in reverse order in the new Fig. 3 B.

  1. 3 A and B. The DNA sequences shown in A and B are different although they claim to present the site of the mutations.

Response to comments:

We are very sorry for the mistake. We have presented the sequence in reverse order in the new Fig. 3 B

  1. 3C. The authors used only one OE line. Do they have additional independent lines to demonstrate that the phenotype results from overexpressing the transgene and not from positional effect of its insertion?

Response to comments:

Many thanks for the comment. OsABA8ox2 overexpressed lines with lower ABA content and faster water loss lead to lower survival rates. Besides, these seeds are intolerant to aging and fail to germinate after aging. There was only one overexpression line survived and used for statistical analysis.

  1. The authors assayed germination but present the germination data in diverse ways: Fig. 1B – ratio (out of 1), Fig. 1E, % (0ut of 100); Fig. 2E, and 4E- survival rather than germination. This is very confusing. Please use a single option. ‘

Response to comments:

Many thanks for the comment. All germinated rice seeds were transferred to soil to calculate the rates of seedling establishment, so there may be a seedling establishment rate higher than the seed germination rate. We have revised the method and fig 1 B.

  1. Line 202. The author used a single ABA concentration but argued that “a certain amount of ABA...”. This is not supported by any data.

Response to comments:

Many thanks for the comment. We have replaced it by “Exogenous”

  1. 4A. The author does not present data for KO-11.

Response to comments:

Many thanks for the comment. We found that two mutants have consistent phenotypes, so only KO-3 was selected for subsequent and repeated experiments.

  1. Seed presented at Fig. 3D and 4D (unaged and aged WT) are the same picture.

Response to comments:

Many thanks for the comment. They are aged and unaged ZH11 seeds, and the photos of multiple treatments are all referenced to it. 

  1. WT is presented as WT (for example Fig. 3C) or ZH11 (for example Fig. 3D). This is very confusing.

 Response to comments:

Many thanks for the comment. We have corrected them with WT in the new Fig 3.

Minor points 

  1. H2O2 should be printed where the digits are in subscripts.

 Response to comments:

Many thanks for the comment. We have corrected them in the new manuscript.

  1. Line 22. As the abstract should be understood by its own, please add the role of DPI.

Response to comments:

Many thanks for the comment. We would define it as “(an H2O2 inhibitor)”

  1. Line 51. Spell out the full name of the PER1 gene.
  2. Line 52. Both species names should be in Italics.

Response to comments 3 to 4:

Many thanks for the comment. We have corrected them as suggested in the new manuscript.

  1. Line 86-7. Please indicate the ratio of embryo (in weight or number) to buffer used.
  2. Line 87. Change “lysate” to lysis buffer.

Response to comments 5 to 6:

Thank you for your suggestions, and we have corrected them in the new manuscript.

  1. Line 88. Did you homogenize the supernatant with H2O2, or mixed it by using a vortex?
  2. Line 88. Please indicate the concentration of the added H2O2 solution used.

Response to comments 7 to 8:

Many thanks for the comment. We are very sorry for the mistake in the confused description. No H2O2 was added during the quantification of H2O2. We have now rephrased our descriptions in supplementary methods.

  1. Line 94. Rephrase to “immersed in 10 mM ABA or 20 mM DPI” 
  2. Line 140 and Figure 1b. The germination rate is presented as %. You show a fraction of germinated seeds (out of 1).

Response to comments 9 to 10:

Thank you for your suggestions, and we have corrected them in the new manuscript.

  1. 2B. The units of H202 (ng embryo-1) are not clear. Is that ng H202 per a single embryo?

Response to comments:

Many thanks for the comment. Yes, each repeat consisted of 25 embryos (about 20 mg) used for the detection of H2O2 content. We have revised the method.

  1. Linev199 – is 1 week (not 1 weeks) after sowing a “immediately following germination”?
  2. 4 D and E. Indicate in the legend what “CK’ is, or change the test under the panel to “ (empty) , +ABA, +DPI”
  3. Please correct styles: species names in Italics. For example, see lines 288 and 299.
  4. References, Check journal names. The first letter should be in CAPS. For example, see line 312 and 317. 

Response to comments 12 to 15:

Thank you for your suggestions, and we have corrected them in the new manuscript.

Reviewer 2 Report

Comments and Suggestions for Authors

Authors demonstrated that ABA inhibits rice seed aging. And authors produced overexpression lines and knockout lines of OsABA8ox2. These lines show interesting phenotypes regarding to seed aging.

H2O2, 2 should be subscript.

 How was the H2O2 contents in OE lines?

 How was the growth rate of KO lines? As shown, KO lines accumulated ABA and ABA may attenuate growth.

Author Response

Reviewer 2

Authors demonstrated that ABA inhibits rice seed aging. And authors produced overexpression lines and knockout lines of OsABA8ox2. These lines show interesting phenotypes regarding to seed aging.

H2O2, 2 should be subscript.

Response to comments:

Thanks for the comment, and we have corrected them in the new manuscript.

How was the H2O2 contents in OE lines?

Response to comments:

Many thanks for the comment. We attempted to measure the H2O2 content in OE lines. However, due to the death of seeds of OE lines during the aging process, no detection was observed.

How was the growth rate of KO lines? As shown, KO lines accumulated ABA and ABA may attenuate growth.

Response to comments:

Many thanks for the comment. Due to the presence of other degrading genes, the ABA content in the KO lines has not reached the level sufficient to inhibit growth. Consequently, the growth of the KO lines remains unaffected.

Reviewer 3 Report

Comments and Suggestions for Authors

see PDF

Comments on the Quality of English Language

Moderate editing (see PDF)

Author Response

Reviewer 3

The draft (Plants-2895059) by Zheng et al (2024) titled “ABA inhibits rice seed aging by reducing H2O2 …” has been assessed. Did you assume that the aging of the seeds wouldn't affect germination? In other words, was it necessary to investigate?

Response to comments:

Many thanks for the comment. It is widely recognized that aging significantly affects seed germination. Therefore, this study aims to elucidate the mechanisms underlying the impact of aging treatments on seed germination and explore methods to enhance seed aging tolerance.

Line 49, ….during….???; line 52, Nymphaea tetragona in A. thaliana; line 53, ..[17]. In rice…; line 53, ...knockout of PER1A..; line 56, ...[18]; . Priming…; line , 59 ... [19];. Nanotechnology…;

Response to comments:

Sorry for the incorrect description. We have revised accordingly.

lines 61-62, this sentence needs clarification “we provide valuable germplasm resources and information for breeding aging-tolerant varieties”. Additionally, a paragraph is needed to clearly define the objectives and conclusions of this work.

Response to comments:

Many thanks for the comment. Yes. We have rephrased our descriptions in the new manuscript.

Comment: The authors provide only a brief overview, comprising six lines (44-49), to establish the "state of the art" necessary to justify the objectives of this work. The introduction is weak overall.

Response to comments:

Yes. Thank you for your suggestions, and we have now more fully described the introduction in the new manuscript.

H2O2 no; H2O2 , yes throughout the entire text. Line 87, 12,000 × g (space); line 96, control (CK) in all text.

Response to comments:

Sorry for the mistake. We have revised it in the new manuscript.

Why is DPI used in the present work?

Response to comments:

Diphenyleneiodonium chloride (DPI), an H2O2 inhibitor. We include a description of DPI in the Abstract part.

Line 105, what does reconstituded mean?

Response to comments:

Many thanks for the comment. “redissolved” would be more accurate. We have revised it in the new manuscript.

Lines 106-107, “...analyzed using an LC-ESI-MS/MS system (consisting of an HPLC Shim-pack UFLC SHI- 106 MADZU CBM30A system coupled with an MS Applied Biosystems 6500 Triple Quadru- 107 pole)...”, I would be unable to repeat this protocol. As this work is conducted by the same group as in Plant J. (2022), these authors should state: the ABA was determined as previously described [21];

Response to comments:

Many thanks for the comment. We have corrected them as suggested in the new manuscript.

line 114, “the seeds were scanned as above” (????).

Response to comments:

Sorry for the confused description. The seeds were scanned using an EPSON V700 Scanner. We have added in the method section.

What is “germination ratio” (Fig. 1B)?? Mat&Met must be slightly corrected.

Response to comments:

Many thanks for the comment. Germination was scored based on a ≥1 mm coleoptile emergence. Germination ratio (%) = number of germinated seeds/total number of seeds × 100%. We have added in the method section.

What does the inside of Fig. D mean?

Response to comments:

Many thanks for the comment. The top right corner of the image shows rice seeds subjected to 48 hours of aging treatment, which fail to develop normal radicles and roots during early seedling growth. Explanation has been added to the figure legend.

How was seedling establisment determined (Fig. E)?

Response to comments:

Many thanks for the comment. Seedling establishment was the percentage of germinated seeds that develop into established seedlings, which is now described in detail in supplementary methods.

Scale bar=1 mm and ±SE (separate) throught tex. Line 158, … agents,. Arrow… Line 165, Wang et al. [8] (italics and separate [8]). When OsABA8ox2 refers to gene, must be written in italics. Line 169, … using the CRISPR/Cas9 method as in [21] ?? Use while and not whereas (line 175).

Response to comments:

Many thanks for the comment. We have corrected them as suggested in the new manuscript.

Fig. 3F: describe the absence of growth in OE-26.

Response to comments:

Thank you for your suggestions. Description has been added to the figure legend.

Line 202 and 220, A certain amount of ABA… Very vague expression.

Response to comments:

Many thanks for the comment. We replaced “A certain amount of” with “Exogenous” in line 202 and “a low concentration of” in line 220 respectively."

Line 203, ...of artificially … in artificially… Line 210, We postulated that ABA reduces …. We postulated that the existence of ABA reduces.

Response to comments:

Thank you for your suggestions. We have corrected them as suggested in the new manuscript.

My doubt is that it is not known whom ABA alters H2O2 levels. That is, who do ABA and H2O2 pathways interrelate in rice?? Perhaps, this subject may be interesting for discussion.

Response to comments:

Thanks for the comment. This is a very good question. We have discussed in the revised Discussion section. The existence of ABA reduces the concentrations of H2O2 during accelerated aging. However, who do ABA and H2O2 pathways interrelate in rice during accelerated seed aging, we are also very interested in this question, which will be research focus in the future study.

Line 222, “this study provides valuable germplasm resources”; to be clearly explained.

Response to comments:

Thanks for the comment. We are sorry for our inappropriate wordings. We have corrected the inappropriate statement to “this study offers novel insights and strategies for breeding and genetic enhancement aimed at improving seed storage longevity and aging tolerance.”

Once corrected as suggested, the results section will be at an acceptable level. Line 237, in addition to [25], include these interesting reviews: [Heredity 2022, 128, 450 – 459; Chinese J. Biotech. 2022, 39, 77-88; The Pharma Inn. J. 2023, 12, 1511-1517; and Antioxidants 2022, 11, 1594].

Response to comments:

Thank you for your suggestions. We have expanded our citation of the literature except “The Pharma Inn. J. 2023, 12, 1511-1517” (We cannot find this work).

This work, although brief, presents interesting results that open up several avenues for discussion. However, the discussion section is too short and insufficient. Acceptance of this draft for publication necessitates further elaboration in the discussion, addressing the weaknesses and issues listed above in a "point by point" manner.

Response to comments:

Thanks for the comments. In response to the reviewer’s comment, we have extended the Discussion section in which we address the important of ABA and H2O2 in aging process and discussed relevant research on the regulation of seed vigor by the ABA signaling pathway, outlining future research directions. Moreover, we provided point-by-point responses to the reviewer's comments.

Round 2

Reviewer 1 Report

Comments and Suggestions for Authors

The authors answered all the points raised in my review

Reviewer 3 Report

Comments and Suggestions for Authors

This new version is already OK